# DAN-SuperPoint: Self-Supervised Feature Point Detection Algorithm with Dual Attention Network

**DOI:** 10.3390/s22051940

**Published:** 2022-03-02

**Authors:** Zhaoyang Li, Jie Cao, Qun Hao, Xue Zhao, Yaqian Ning, Dongxing Li

**Affiliations:** 1School of Mechanical Engineering, Shandong University of Technology, Zibo 255000, China; 20501020029@stumail.sdut.edu.cn (Z.L.); lidongxing@sdut.edu.cn (D.L.); 2School of Optics and Photonics, Beijing Institute of Technology, Beijing 100081, China; caojie@bit.edu.cn (J.C.); 3220210544@bit.edu.cn (X.Z.); ningyq@bit.edu.cn (Y.N.)

**Keywords:** feature point detection, attention module, multi-scale feature fusion, deep learning

## Abstract

In view of the poor performance of traditional feature point detection methods in low-texture situations, we design a new self-supervised feature extraction network that can be applied to the visual odometer (VO) front-end feature extraction module based on the deep learning method. First, the network uses the feature pyramid structure to perform multi-scale feature fusion to obtain a feature map containing multi-scale information. Then, the feature map is passed through the position attention module and the channel attention module to obtain the feature dependency relationship of the spatial dimension and the channel dimension, respectively, and the weighted spatial feature map and the channel feature map are added element by element to enhance the feature representation. Finally, the weighted feature maps are trained for detectors and descriptors respectively. In addition, in order to improve the prediction accuracy of feature point locations and speed up the network convergence, we add a confidence loss term and a tolerance loss term to the loss functions of the detector and descriptor, respectively. The experiments show that our network achieves satisfactory performance under the Hpatches dataset and KITTI dataset, indicating the reliability of the network.

## 1. Introduction

The detection of feature points and the establishment of descriptors are important steps in image matching. In computer vision-based applications such as simultaneous localization and mapping (SLAM), structure-from-motion (SFM), and image retrieval, the processing of image feature points determines the correspondence between different images. Accurate extraction of feature points can improve the matching accuracy of images. With the wide applications of computer vision and the more complex environment faced by image processing, it is particularly important to find a stable feature point detection method.

At present, the processing methods for image feature points can be divided into traditional methods and deep learning-based methods. Traditional feature extraction methods are difficult to achieve satisfactory performance in challenging situations. The scale invariant feature transform (SIFT) algorithm [1] was scale invariant but not real-time. Rubele et al. [2] proposed the oriented fast and rotated brief (ORB) algorithm, which was improved on the basis of the features from accelerated segment test (FAST) algorithm [3] to make the feature points have rotation invariance and real-time performance. Mair et al. [4] proposed the adaptive and generic corner detection based on the accelerated segment test (AGAST) algorithm, which can maintain consistent angular responses without training and has the same reusability as the FAST algorithm. However, the above algorithms cannot extract a sufficient number of feature points in low texture scenes and cannot keep the accuracy of feature point extraction stable. Samuele [5] proposed a feature point detection method based on the wave equation, which can maintain a certain accuracy on low-texture objects with symmetry, but was not suitable for irregular scenes with large changes. The feature point detection method based on deep learning still has high stability in a low texture environment. [6] proposed a novel deep network structure for end-to-end differentiability. It can realize the complete feature point processing process, but the network structure was complex. [7] proposed a learning-based method to detect repeated feature points, which can still maintain good accuracy under the challenge of complex environments, but the detection of feature points was not extended to the scale space. References [8,9,10] proposed different network structures for feature point detection. Among them, the feature maps used to detect feature points in [8,10] lose part of the information. Although the feature maps used for feature point detection in [9] were rich in information, the network structure was complex and cannot meet the requirements of real-time performance. In feature extraction, the balance between accuracy and real-time performance of deep learning-based methods has always been a focal issue.

Aiming at the above issue, we propose a deep learning-based self-supervised feature point detection network with an attention mechanism that can be applied to feature extraction modules in VO. First, the feature pyramid networks (FPN) [11] is used to extract feature maps of different scales for multi-scale feature fusion. Then, the obtained feature maps through the spatial attention module and the channel attention module [12] to establish spatial correspondence and channel correspondence, respectively. Finally, the weighted feature maps are output for the training of detectors and descriptors. Our method can replace traditional feature point detectors in VO, taking advantage of the high stability of deep learning to improve the accuracy of the system. In addition, we design loss functions for detector and descriptor training. In the detector head, we add softargmax to improve the prediction accuracy of feature points and add a confidence loss term to ensure the reliability of feature points. We add a prediction tolerance loss term based on dense feature descriptors to speed up the convergence of the network. The feature descriptor generated by the network is the same as the descriptor format of the ORB_SLAM2 system [13], and the network can be directly applied to the SLAM system instead of the original feature extraction module.

The rest of the article is organized as follows. Section 2 is a review of related work. Section 3 introduces our designed network structure and loss function. Section 4 is the experimental results and analysis of our network on different datasets after training. Section 5 is the discussion part of the article. Section 6 is the conclusion of the article.

## 2. Related Work

### 2.1. Local Feature Learning

The feature point detection method based on deep learning improved the stability of feature point detection. Due to the unclear definition of feature points, adding effective labels to images became a difficulty in the detector training process. Therefore, most methods only work on local descriptors of image patches [14,15]. However, Quad-networks [16] used an unsupervised learning two-layer neural network on patches to learn to define a good feature point to effectively address this issue, but did not provide corresponding descriptors for each patch. LF-Net [17] is an end-to-end differentiable network similar to [6]. It can quickly learn on full images but did not share computation during the training of network feature points and the use of image patches limits the network’s training of descriptors. Geometric correspondence network (GCN) [9] combined convolutional neural networks (CNN) and recurrent neural networks (RNN) for detector and descriptor training. It had better motion estimation compared to related deep learning methods and handcrafted methods. However, the network structure is too large, which made the network computation heavy and cannot meet the requirement of real-time performance. GCNv2 [10] proposed a network that can run on low-performance devices. However, it cannot improve itself online. The self-supervised framework of SuperPoint [8] can effectively solve this issue, similar to [18] using a self-supervised way to generate a synthetic dataset for training, and obtained the same comparable results with the SIFT method. Li et al. [19] proposed a multi-task framework for training feature point detectors and descriptors for the complex network structure of [8,9]. It sacrificed some precision while increasing speed. UnsuperPoint [20] improved on [8] so that the network only needs one round of training to meet the prediction requirements and does not need to generate labels for ground truth points, but did not embed the network into SLAM for testing. The feature detection network proposed by [21] can directly optimize the geometric pose targets after completing feature extraction and has better generalization ability to unknown datasets compared to existing learning-based methods. However, the network framework integrates multiple modules, which put forward higher requirements on computing power compared with other methods. Luo et al. [22] proposed a network for precise shape localization using D2-Net [23] as the backbone structure to learn detector and descriptor. It improved localization accuracy through three lightweight but effective optimizations. R2D2 [24] jointly learned feature point detector and descriptor in high-confidence regions. It effectively avoided the effect of ambiguous regions to enhance the detection and description of feature points. In addition, in order to further improve the accuracy of feature point detection, Key.Net [25] combined hand-made filters with learned filters and determine the search range of feature points through a multi-scale index layer. It significantly reduced the complexity of learnable parameters and detector structure.

### 2.2. The Combination of Feature Extraction and Attention Mechanism

The combination of deep learning-based feature point detection methods and attention mechanisms had achieved good results. Non-local neural networks [26] enable a single feature at any location to capture long-range dependencies through a self-attention mechanism to perceive contextual information. It can be combined with existing architectures to significantly improve network accuracy. The successful application of Transformer [27] in various natural language processing (NLP) tasks inspired scholars to explore computer vision tasks. A two-stage local image feature matching method based on transform (LoFTP) [28] had great advantages on public datasets. The combination of coarse pixel-level dense matching and fine refined matching enables the method to still produce dense matching in low-texture regions. However, the two-stage processing method increased processing time while maintaining accuracy. Wang et al. [29] proposed a new soft point-wise transformer model (SPTD2) for the training of descriptor and detector. The model focused on the intrinsic correlation and multi-scale correlation of local features, which was advantageous for high-resolution feature mapping. Although the model reduced the computational complexity and GPU memory compared to the original attention module [12,26,30,31], its memory was still too large at hundreds of megabytes.

In this work, we train in a self-supervised manner and add a dual attention network. First, the feature map is obtained through the FPN structure. Then, feature dependencies are captured through a spatial attention mechanism and a channel attention mechanism. Finally, the detectors and descriptors are trained using the weighted feature maps. The descriptor output by our network is 256-bit, which is the same as the traditional ORB feature descriptor format and can be easily replaced.

## 3. Method

We design a self-supervised feature extraction network with a dual attention mechanism. The network framework adopts the current mainstream feature extraction network structure, as shown in Figure 1.

The network is designed as a common encoder, feature pyramid structure, dual attention network, and two different decoders. Among them, the feature map P_2_ is weighted by the dual attention network to obtain the feature map P_3_. The network can not only realize the sharing of parameters at runtime but also realize the simultaneous operation of multiple tasks by different decoders working at the same time.

### 3.1. Network Structure

BackBone: The backbone of the network adopts the VGG [32] structure, which consists of blocks with the same structure. Each block contains two convolutional layers, bn layers, and nonlinear activation functions. Where each block uses 3×3 convolution kernel as shown in Figure 2a. The size of our commonly used convolution kernel is shown in Figure 2. The 3×3 convolution kernel is the smallest size capable of capturing pixel-eight neighborhood information. Although a large convolutional kernel can achieve a large receptive field, we can achieve the same receptive field by stacking small convolutional layers to replace large convolutional layers. More importantly, stacking multiple convolutional layers has more nonlinearities and fewer parameters than one large convolutional layer. In order to ensure the efficiency of the network operation, we use a 6-layer convolutional structure, and the number of channels of the convolutional layer is set to 32-32-64-64-128-128. The input image is $I^{H\times W\times 1}$, and the multi-scale feature map F is obtained by convolution operation on the input image.

Feature pyramid structure: First, we reduce the dimension of feature map F3 to 64 channels and obtain feature map $I_{in}^{H_{s}\times W_{s}\times 64}$ through bilinear interpolation. Then, the multi-scale feature fusion of the image is performed to obtain the feature map $P_{1}^{H_{s}\times W_{s}\times 64}$. Finally, the feature map $P_{1}^{H_{s}\times W_{s}\times 64}$ is convolved again to obtain a feature map $P_{2}^{H_{z}\times W_{z}\times 128}$. We use $P_{2}^{H_{z}\times W_{z}\times 128}$ for subsequent decoding work.

Attention network: In order to enhance the global features of the feature maps after multi-scale fusion, we model the dependencies on the spatial and channel dimensions of $P_{2}^{H_{z}\times W_{z}\times 128}$ through the location attention module and the channel attention module, respectively. The location attention module determines the spatial dependencies between any two locations. The channel attention module captures the channel dependencies between any two-channel maps and updates them using the weighting of all channel maps. The outputs of the two attention modules are fused to enhance the feature representation.

Detector head: Conv1 contains two 3×3 convolutional layers. The second layer is a dimensionality reduction layer with a stride of 1. First, the feature map $P_{3}^{H_{z}\times W_{z}\times 128}$ is converted into a feature map $I_{z}^{H_{z}\times W_{z}\times 65}$ with 65 channels after Conv1 operation. We get the coordinates X_Position_ and Y_Position_ of the feature points through $I_{z}^{H_{z}\times W_{z}\times 65}$. Then, we adjust the resolution back to the original image by upsampling through the Reshape operation to obtain $I_{se}^{H\times W\times 1}$. Finally, the calculation of feature points is performed on the full-resolution map $I_{se}^{H\times W\times 1}$.

Descriptor head: Similar to the processing process of the detector, the input feature map $P_{3}^{H_{z}\times W_{z}\times 128}$ needs to be output through the Conv2 operation. Conv2 contains two convolutional layers. The parameters of the first layer are the same as those of Conv1. The second layer is still a dimensionality reduction layer, but the number of output channels is set to 256, which is also to facilitate subsequent transplant operations. The feature map $I_{dc}^{H_{z}\times W_{z}\times 256}$ output by Conv2 is first restored to full resolution by bilinear interpolation. Then, normalized to unit length by L2-Norm. Finally, output the feature map $I_{de}^{H\times W\times 256}$ with the same size as the original image and the number of channels is 256.

### 3.2. Dual Attention Network Weights

The location attention module adds contextual information to local features to enhance their representation. As shown in Figure 1, the fused feature map $P_{2}^{H_{z}\times W_{z}\times 128}$ needs to undergo a dimensionality reduction operation in the attention module to obtain $O_{}{}^{H_{sa}\times W_{sa}\times C_{sa}}$. $O_{}{}^{H_{sa}\times W_{sa}\times C_{sa}}$ is processed by the convolution layer to obtain three mappings of B, C, and D, $\left\{B,C,D\right\}\in R^{H_{sa}\times W_{sa}\times C_{sa}}$. We multiply C and the transposed matrix of B and compute the spatial attention map A through the softmax layer. We perform a matrix multiplication operation of D with the transpose of A, and then perform an element-wise addition with O. The calculation process of spatial attention map and output feature map is as follows:(1)$$a_{mn}=\frac{\exp\left(B_{n}\cdot C_{m}\right)}{\sum_{n=1}^{N}\exp\left(B_{n}\cdot C_{m}\right)},A\in R^{G\times G}$$
(2)$$J_{m}=\alpha_{sa}\sum_{n=1}^{N}(a_{mn}D_{n})+O_{m}$$
where $a_{mn}$ represents the correlation between the $n^{th}$ position and the $m^{th}$ position, and the larger the value, the greater the similarity, $\alpha_{sa}$ represents the scaling factor, $G=H_{sa}\times W_{sa}$.

The channel attention module improves feature representation by establishing interdependencies between channels. Unlike the positional attention module, the channel attention module utilizes the feature map $O_{}{}^{H_{sa}\times W_{sa}\times C_{sa}}$ to be multiplied by its own transposed matrix. Then, the channel attention map $K\in R^{C\times C}$ is obtained through the softmax layer. Finally, we perform a matrix multiplication operation of K with the transpose of O, and then perform an element-wise addition with O. The calculation process is as follows:(3)$$k_{mn}=\frac{\exp\left(O_{n}\cdot O_{m}\right)}{\sum_{n=1}^{C}\exp\left(O_{n}\cdot O_{m}\right)}$$
(4)$$J_{m}={℘}_{sa}\sum_{n=1}^{C}(k_{mn}O_{n})+O_{m}$$
where $k_{mn}$ represents the correlation between the $n^{th}$ channel and the $m^{th}$ channel. ${℘}_{sa}$ represents the scaling factor.

### 3.3. Loss Functions

The overall loss computation consists of the detector loss $L_{f}$ and the descriptor loss $L_{d}$, as shown in (5). We adopt a training form that optimizes both parts of the loss simultaneously. In order to balance the two-part loss, we also set two additional weight parameters to ensure the correct convergence of the loss function.
(5)$$L_{sum}=\alpha_{f}L_{f}+\alpha_{d}L_{d}$$
where $L_{f}$ is the detector loss function, $L_{d}$ is the descriptor loss function, $\alpha_{f}$ is the weight coefficient of $L_{f}$, and $\alpha_{d}$ is the weight coefficient of $L_{d}$.

#### 3.3.1. Detector Head Loss

The loss calculation of the detector is divided into two parts. The first part is to calculate the scores of the two frames of images after the homography transformation. The second part is to calculate the error and confidence between the predicted value of the feature point and the true value. The loss of the detector is calculated as follows:(6)$$L_{f}=L_{ft}(x_{or},y_{or})+L_{ft}(x_{wr},y_{wr})+L_{fa}(x_{pr},y_{pr})+L_{fa}(x_{pw},y_{pw})$$
where $L_{ft}$ represents the score loss, and $L_{fa}$ represents the position error loss of the feature points. $\left(x_{or},y_{or}\right)$ represents the original image, $\left(x_{wr},y_{wr}\right)$ represents the transformed image, $\left(x_{pr},y_{pr}\right)$ represents the predicted value of the original image, and $\left(x_{pw},y_{pw}\right)$ represents the transformed predicted value.

Feature point detection can be regarded as a binary classification problem. Most of the previous work is based on the cross-entropy loss function. However, it is easy to deviate when the sample distribution is not balanced. Therefore, we apply the loss function focal loss [33] during model training to improve the convergence speed in feature point detection. The loss is calculated as follows:(7)$$L_{ft}(x,y)=\frac{1}{U_{\omega }V_{\omega }}\sum_{\begin{array}{l} u=1\\ v=1 \end{array}}^{U_{\omega },V_{\omega }}F_{ft},\;x\in I_{z}^{H_{z}\times W_{z}\times 65}$$
(8)$$PF(x_{uv};y)=\frac{\exp\left(x_{uvy}\right)}{\sum_{d=1}^{65}\exp\left(x_{uvd}\right)}$$
(9)$$F_{ft}=-\alpha_{\theta }{\left(1-PF\right)}^{\lambda }\log\left(PF\right)$$
where Uω=H8, Vω=W8, y is the label of the ground truth value of the feature point, $\alpha_{\theta }$ is used to suppress the unbalanced number of positive and negative samples, and λ is used to control the unbalanced number of difficult and easy samples.

In order to further improve the estimation accuracy of feature points, we are inspired by the literature [21], adding the softargmax function to the patch of 5×5 in the neighborhood of each feature point to further refine the coordinate position of each feature point, and update the coordinates with sub-pixel accuracy. The calculation process is as follows:(10)$$T_{pi}=T_{o}+\left(\Delta x,\Delta y\right)$$
(11)$$\Delta x=\frac{\sum_{i}\sum_{j}e^{T_{g}}j}{\sum_{i}\sum_{j}e^{T_{g}}}\;,\;\Delta y=\frac{\sum_{i}\sum_{j}e^{T_{g}}j}{\sum_{i}\sum_{j}e^{T_{g}}}$$
(12)$$L_{fat}={\left\|T_{pi}-T_{t}\right\|}_{2}$$
where $T_{pi}$ represents the updated predicted value coordinates, $T_{o}$ represents the center pixel coordinate value, $T_{g}$ represents the pixel value at the position of the heat map g, and $T_{t}$ represents the true value.

The error loss of the feature point coordinate value includes error accumulation, error mean and confidence. The error accumulation part is to ensure the overall accuracy of feature point prediction. The error mean and the confidence loss term are to ensure the stability of the accuracy of the extracted feature points of the trained network.
(13)$$L_{fa}=L_{fat1}+L_{fat2}+L_{fat3}$$
(14)$$L_{fa1}=\zeta \log\left(\sum_{i=1}^{N}L_{fat}\right)$$
(15)$$L_{fa2}=\frac{\sum_{pi=1}^{N}L_{fat}}{N}$$
(16)$$L_{fa3}=\phi \sum \left(L_{fat}-L_{fa2}\right)$$
where $L_{fa1}$ represents the cumulative sum of errors, ζ is the weight coefficient, $L_{fa2}$ represents the mean value of the error, $L_{fa3}$ represents the confidence, and ϕ is the weight coefficient.

#### 3.3.2. Descriptor Head Loss

The two image frames for which the descriptor loss is calculated are the image pairs after homography transformation. We obtain the final descriptor loss by computing the homography corresponding point pairs between the two images. At the same time, in order to improve the network training accuracy and the network convergence speed, we set the error loss term of the descriptor. The descriptor from the original image is $d\in I_{}^{H_{z}\times W_{z}\times 256}$, and the descriptor in the transformed image is $d_{\hbar }\in I_{\hbar }^{H_{z}\times W_{z}\times 256}$. We set the matching threshold S to 4-pixel values during training. The specific calculation process is as follows:(17)$$Z=\left\{\begin{array}{ll} 1, & if\;M\leq S\;\\ 0, & otherwise \end{array}\right.$$
(18)$$M=\left\|N-N_{\hbar }\right\|$$
(19)$$L={\left\|N-N_{\hbar }\right\|}_{2}$$
where M represents the interval between two pixels, N represents the coordinates of the center coordinates of a pixel in a unit after homography transformation, $N_{\hbar }$ is the center pixel coordinates of the corresponding unit in the image after homography transformation, L Represents the 2 norm of the spacing between pixels.

Since the number of corresponding points between frames is significantly less than the number of non-corresponding points, we reduce the impact of the high loss of non-corresponding points on the overall descriptor loss function by setting a modulation factor ℑ. At the same time, a hinge loss function is applied to add upper and lower bounds for prediction. $r_{p}$ and $r_{n}$ are the upper and lower boundaries, respectively.
(20)$$L_{d}=\frac{1}{{\left(KK_{\hbar }\right)}^{2}}\sum_{k=1}^{K}\sum_{k_{\hbar }=1}^{K_{\hbar }}F_{m}+\psi \sum L$$
(21)$$F_{m}=ℑ\cdot Z\cdot \max(0,r_{p}-d^{Τ}d_{\hbar })+(1-Z)\cdot \max(0,d^{Τ}d_{\hbar }-r_{n})$$
where K and $K_{\hbar }$ represent the number of corresponding points and non-corresponding points in the original image and the two frames of images after homography transformation, respectively. ψ is the weight coefficient of the descriptor cumulative error loss term.

## 4. Experiments and Analysis

Our experiments are carried out under the pytorch [34] framework. The network training is divided into two steps. The first training is 200,000 iterations on the synthetic dataset. The second training is 200,000 times on the COCO dataset [35] or KITTI dataset [36] annotated with the network parameters of the first training. Then, it is tested with the trained network model under the Hpatches dataset [37]. Finally, a total of 10 image sequences from 01 to 10 under the Odometry dataset provided by the KITTI dataset are used in VO to test the effect of the proposed algorithm. We perform data augmentation on the training data to improve the robustness of the network to illumination and pilot changes.

The descriptor size is set to 256 bits in our experiments. We found that the descriptor loss is much larger than the detector loss during network training. Therefore, we balance the two-part loss by the weight parameter to make the network converge correctly. First, we set the initial values to $\alpha_{f}$ = 1, $\alpha_{d}$ = 0.1. Then, we resize $\alpha_{d}$ by a factor of 10. Finally, we set the $\alpha_{f}$ and $\alpha_{d}$ value to $\alpha_{f}$ = 1, $\alpha_{d}$ = 0.0001, respectively, through pre-training debugging and referring to the setting of this parameter in [8,21]. During the training of the network, $L_{fa2}$ determines the overall accuracy of the detector, while $L_{fa1}$ and $L_{fa3}$ are further optimizations for the detector. Therefore, we set the appropriate initial values of ζ and ϕ after determining the magnitude of $L_{fa1}$ and $L_{fa3}$ by a simple calculation. Then, we set ζ = 0.0001 and ϕ = 0.001 through the pre-training debugging method. According to the application environment and [33], we set the parameters of the focal loss function to $\alpha_{\theta }$ = 0.25, λ = 2. According to the experience of [8,19] and our training results, the relevant parameters in the descriptor loss function are adjusted. We set ℑ = 250 to keep the descriptors balanced. We guarantee the accuracy of descriptor learning by setting a positive threshold $r_{p}$ = 1 and a negative threshold $r_{n}$ = 0.2. We set ψ = 0.0001 to ensure the convergence speed of the network. When the value of ψ is increased, the network training effect is poor. Conversely, when decreasing the value of ψ, this term has little effect on the network. To ensure the convergence of the loss function, we set the learning rate of the ADAM optimizer to 0.0001. The system used in our experiment is Ubuntu18.04, the CPU is Intel®CoreTM i9-10980XE produced by Intel Corporation of America, the GPU is NVIDIA RTX3080TI produced by NVIDIA in the United States, and the memory is 128 GB.

### 4.1. Feature Point Detection and Matching

The ability of our network and other algorithms to extract feature points in different environments is tested on the Hpatches dataset. Figure 3 shows the effect of applying different algorithms to extract feature points and complete matching in the image pair after homography transformation. The red line is the wrong match. It can be seen from Figure 3 that the network we trained has the best matching effect in the three sets of experiments. Although our network has more false matches in some images than SuperPoint algorithm, our network has more matches. The SIFT algorithm has the largest number of matches, but it has a large number of false matches, which will reduce the system accuracy in subsequent pose estimation. Therefore, our network performance is reliable.

The data in Table 1 are obtained by computing 1000 feature points at 480×640 resolution. It can be seen from Table 1 that our network homography estimation score is the highest when the tolerance threshold is 1 and 3, and slightly lower than SuperPoint when the tolerance threshold is 5. Among them, the ORB algorithm has the highest repeatability of feature points, but the matching effect of homography estimation is poor. It is worth noting that our network is much higher than other algorithms at a threshold of 1, which indicates that our network performs better in pixel localization accuracy. Figure 4 shows the test results of the feature point detection ability of different algorithms under the Hpatches dataset. Combining the repeatability of the feature points with the matching score estimated by the homography, our network performs the best.

We tested the running time of different algorithms under Opencv. Table 1 shows the running speed of different algorithms to extract 1000 points on an image with a resolution of 1242×376. Due to performing multi-scale fusion of feature maps and adding a dual-channel attention module to improve the accuracy of network feature extraction, our network is slower than other algorithms. We improve accuracy at the expense of some time.

### 4.2. KITTI Dataset Test

We select the 01~10 image sequence with ground truth values from the odometry dataset under the KITTI dataset for testing. The feature point extraction ability and the estimation accuracy of the camera trajectory of our network and the three traditional algorithms of SIFT, ORB, and FAST and the deep learning based SuperPoint algorithm are evaluated in different scenarios.

Figure 5 is a screenshot of the five algorithms in different sequences when tested under the VO framework. As can be seen from Figure 5, the number of feature points detected by our network on shadowed parts and irregular objects such as flowers, plants, and trees significantly exceeds that of other algorithms. We applied non-maximum suppression during the training of the network to make the extracted feature points evenly distributed. On the contrary, because the ORB algorithm does not apply non-maximum suppression, the feature points are clustered obviously, which also affects the overall tracking effect. SIFT and FAST algorithms have good feature point detection performance in general scenarios. However, they do not perform as well as learning-based algorithms in poor lighting conditions. Compared with the Superpoint algorithm, our network extracts more feature points and performs more stable in low-texture scenes.

Figure 6 shows the complete camera trajectories estimated by different algorithms under two different sequences of the KITTI dataset, where Our-VO represents our network and Ground Truth represents the ground truth of the camera trajectory. As can be seen from the trajectory in Figure 6, the complete camera trajectory we estimated fits best with the true value of the camera trajectory, and the accuracy of our estimated camera trajectory is significantly improved compared with other algorithms.

Figure 7 is a graph of absolute trajectory error curves of different algorithms under different sequences. From Figure 7, we can clearly see that our estimated camera trajectory has the smallest error compared to others. Compared to other algorithms in terms of accuracy and stability, our network achieves satisfactory results.

Figure 8 shows the number of feature points extracted by different algorithms under the 04 sequence. It can be seen from the graph in Figure 8 that the feature extraction stability of the learning-based method is higher than that of the classical method, and the number of features extracted on each frame of the entire image sequence fluctuates less. The number of features extracted by us exceeds the SuperPoint algorithm while ensuring stability. In addition, it is worth noting that the number of features extracted by the ORB algorithm and the SIFT algorithm fluctuates significantly. This is because they have a stronger ability to extract feature points when the complexity of the environmental conditions is small, so the number of extracted feature points is higher than ours. However, they extract a small number of feature points in low-texture scenes, and deep learning-based algorithms have a stable ability to extract feature points even in low-texture scenes. Looking at the overall curve, our network is more stable than other algorithms. Besides, we have a stronger ability to extract feature points compared to them.

Table 2 is the absolute trajectory error data of different algorithms under different sequences under the KITTI dataset. The data in Table 2 show that our network performs the best overall among the 10 sequences. Among them, compared with SuperPoint and FAST in the 03 sequence, the error of our network is reduced by 80.13% and 79.78%, respectively. Compared with the improved SuperPoint algorithm PC-SuperPoint algorithm [38] in the 04 sequence, the error of our network is reduced by 70.87%, and the error is reduced by 74.93% compared with the best performing SIFT algorithm in the traditional algorithm. In the 05, 08, 09, and 10 sequences, the errors of our network are significantly reduced compared with other algorithms. Although our network is not the best in some sequences, it is the most stable by the error performance of all sequences.

Table 3 shows the relative trajectory error performance of different algorithms under different sequences. The relative trajectory error of camera trajectories estimated by our network is lower than other algorithms in most sequences. Even though it is not the best in the partial sequence, our error is small compared to other algorithms. The data in Table 3 demonstrate that our network has high stability in estimating camera poses.

## 5. Discussion

Figure 3 and Figure 4, and Table 1 are the experimental results of our network and other algorithms under the Hpatches dataset. Compared with traditional algorithms and related deep learning-based algorithms, our network has a significant improvement in accuracy due to the addition of FPN and attention modules to optimize feature maps. However, the increase in processing increases the processing time of the network accordingly. In terms of time, our network is not dominant. Figure 5, Figure 6, Figure 7 and Figure 8 and Table 2 and Table 3 are the qualitative and quantitative analysis of our network and other algorithms under the KITTI dataset. In the VO test, our network meets the requirements for camera tracking in all 10 image sequences under the KITTI dataset. In addition, the test results in the 03, 04, 05, 08, 09, 10 sequences are significantly improved compared with other algorithms. The lighting conditions and environments in different image sequences are different, which puts forward higher requirements for the detection algorithm of feature points. The feature point detection capability of deep learning-based algorithms is more stable than traditional algorithms. Therefore, it can still maintain high detection accuracy in scenes with complex environments and poor lighting conditions. The number of extracted feature points is stable during the tracking process of the entire image sequence. The traditional algorithm has a strong ability to extract feature points when the detection conditions are good, but is poor when the detection conditions are poor, which leads to the large fluctuation in Figure 8. The optimization of feature maps by our network has a stronger detection ability than other learning-based algorithms. Therefore, both the detection number of feature points and the tracking of camera trajectories have achieved satisfactory results.

It should be noted that we improve the detection accuracy by refining the processing of feature maps while increasing the computational time of the network. Although we take into account both accuracy and efficiency when designing the network, we still have a gap compared with traditional feature point detection methods.

## 6. Conclusions

In this paper, we propose a new self-supervised feature extraction network to address the poor performance of traditional feature point detection methods in low-texture situations. First, we achieve feature map fusion through the FPN to enhance the multi-scale detection of the network. Then, in order to improve the accuracy of feature point detection in the network, we propose to use a dual attention mechanism to achieve spatial weighting and channel weighting for a fused feature map. Finally, we set the descriptor output channel to 256 to facilitate subsequent transplantation and design the loss function to improve the prediction accuracy of feature point locations and speed up network convergence. Experimental results show that our proposed network is effective and accurate.

In future work, our research direction is to further optimize the network structure to improve the real-time performance of the network under the condition of ensuring accuracy.

## Figures and Tables

**Figure 1 sensors-22-01940-f001:**
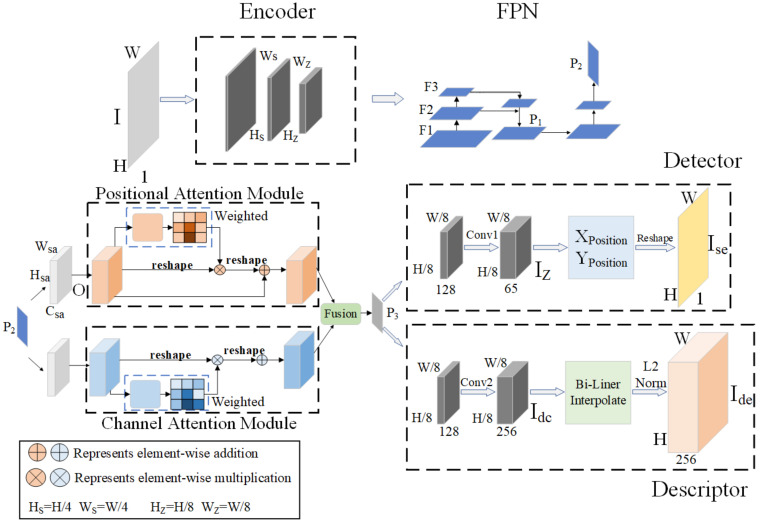
The overall framework of the network. I has been processed by the Encoder, FPN, Position Attention Module, and Channel Attention Module successively, and then the training is completed by decoding the Detector and the Descriptor, respectively. H and W represent the height and width of the image, respectively. H_S_, W_S_, H_Z_, and W_Z_ represent the height and width of different feature maps in the encoding process, respectively. I represents the input image. F1, F2, and F3 are feature maps of different sizes output by the encoder. The feature map P_2_ is obtained after convolution of the feature map P_1_ fused by F2 and F3. W_sa_, H_sa_, C_sa_ are the width, height, and channel of the input feature map O of the attention mechanism. The feature map P_2_ is weighted by the attention network to obtain the feature map P_3_. I_Z_ and I_dc_ represent the feature maps output after Conv1 and Conv2 operations, respectively. I_se_ and I_de_ represent the feature maps output by the detector and descriptor after decoding.

**Figure 2 sensors-22-01940-f002:**
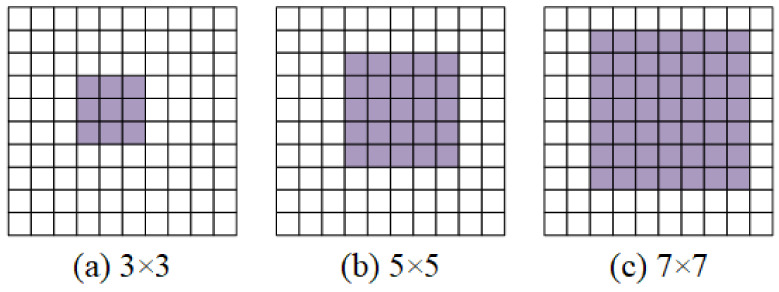
Processing of different convolution kernels. The purple area in the figure is the size of the convolution kernel. (**a**) represents a 3 × 3 convolution kernel. (**b**) represents a 5 × 5 convolution kernel. (**c**) represents a 7 × 7 convolution kernel.

**Figure 3 sensors-22-01940-f003:**
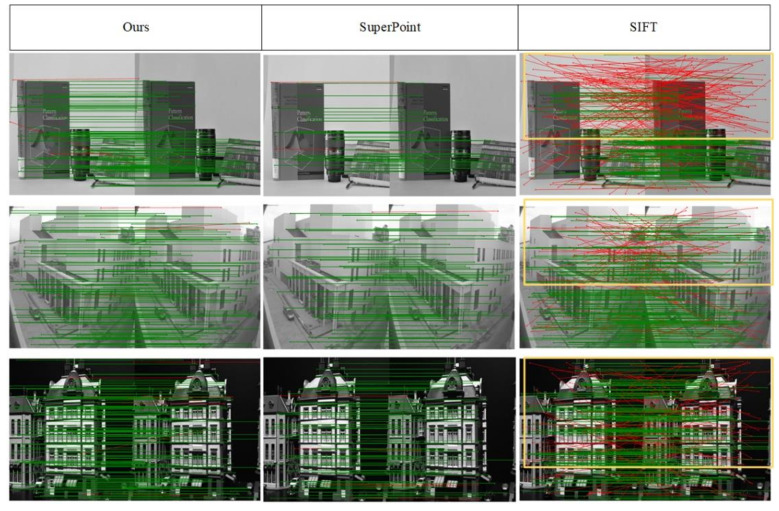
Matching performance of different algorithms. The red lines represent incorrect results. The green lines represent the correct results. We marked a large number of incorrect results with yellow boxes in the results of the SIFT algorithm. The SIFT algorithm performs poorly compared to learning-based methods. Compared with superpoint, our method can produce denser matches with guaranteed matching accuracy.

**Figure 4 sensors-22-01940-f004:**
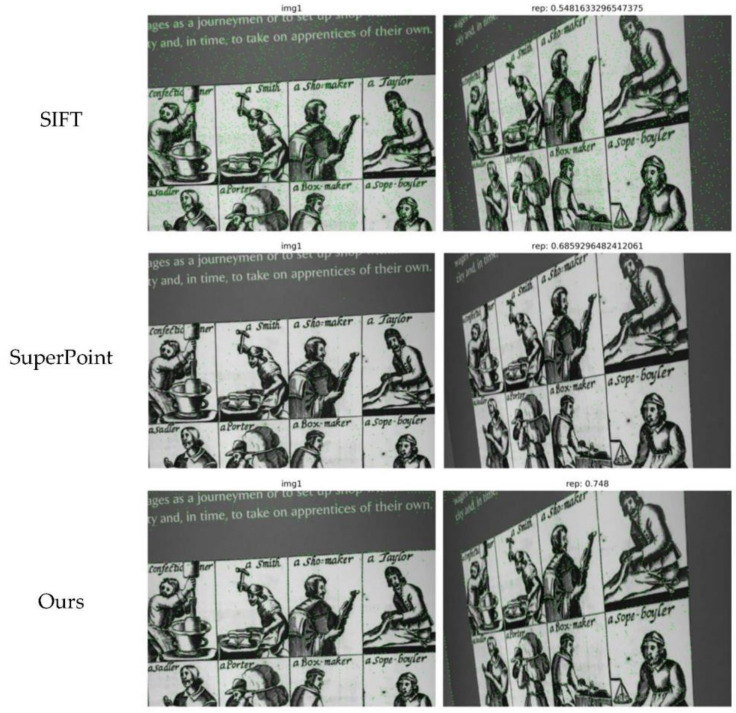
Feature point repeatability detection. On the left is the original image. On the right is the image after applying the homography transformation. rep is the repeatability of feature points.

**Figure 5 sensors-22-01940-f005:**
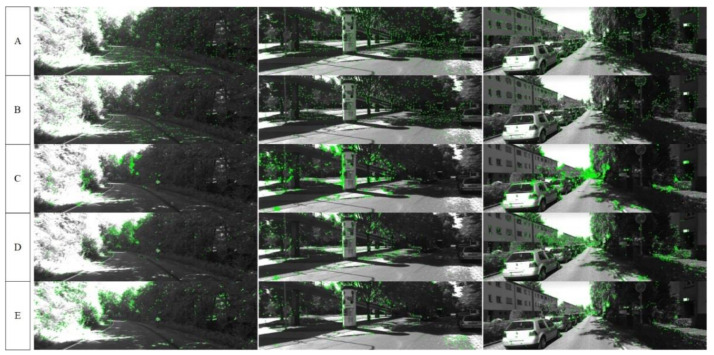
Feature points extracted by different algorithms in the Kitti dataset. (**A**) represents ours, (**B**) represents SuperPoint, (**C**) represents ORB, (**D**) represents SIFT, (**E**) represents FAST.

**Figure 6 sensors-22-01940-f006:**
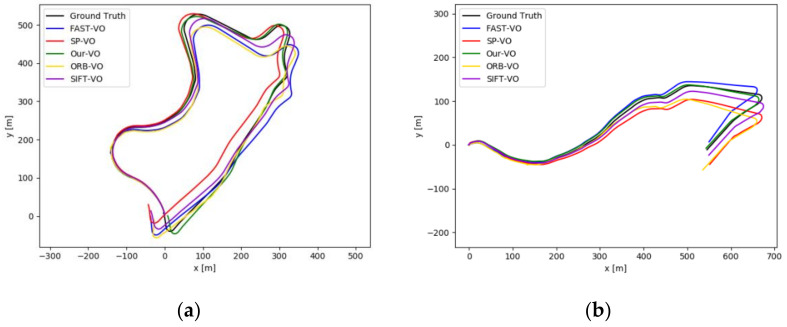
Camera trajectory. (**a**) represents the 09 sequence. (**b**) represents the 10 sequence.

**Figure 7 sensors-22-01940-f007:**
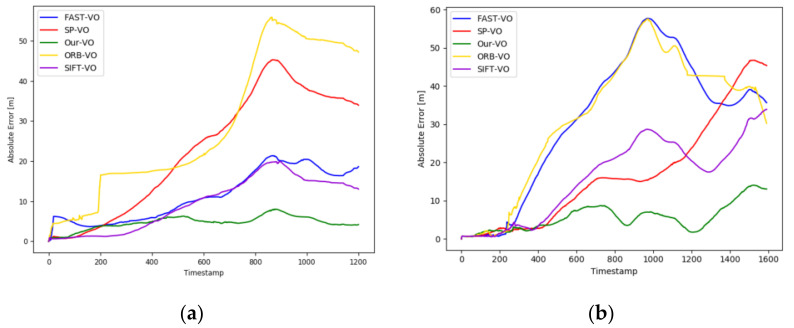
Absolute trajectory error. (**a**) represents the 09 sequence. (**b**) represents the 10 sequence.

**Figure 8 sensors-22-01940-f008:**
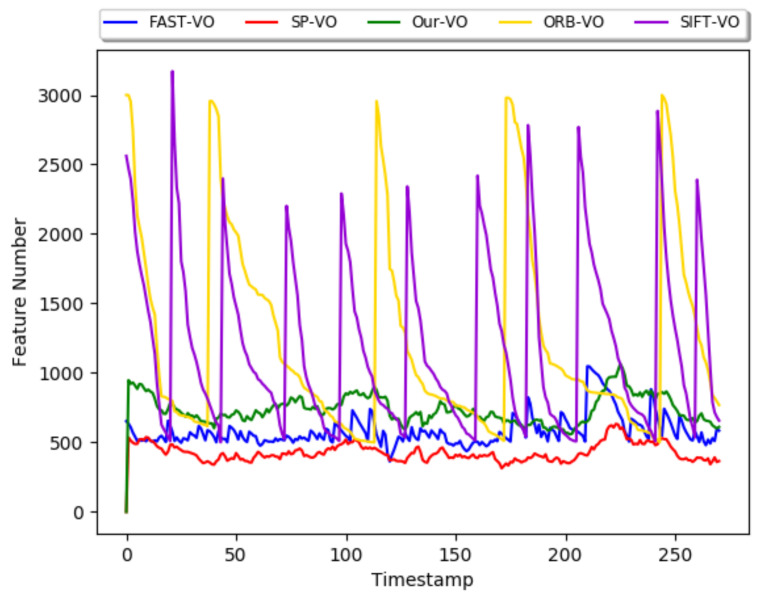
Number of extracted features.

**Table 1 sensors-22-01940-t001:** Homography estimation.

	HomoGraphy Estimation			Repeatability	Time (ms)
	Epsilon = 1	3	5		
Superpoint	0.331	0.684	0.829	0.581	103
LIFT	0.284	0.598	0.717	0.449	
SIFT	0.424	0.676	0.759	0.495	80
ORB	0.150	0.395	0.538	0.641	125
BRISK	0.300	0.653	0.746	0.566	
Ours	0.505	0.729	0.788	0.586	170

**Table 2 sensors-22-01940-t002:** Absolute trajectory errors of different algorithms in different sequences.

Dataset	Ours	SuperPoint	ORB	SIFT	FAST	PC-SuperPoint
01	88.082	85.587	875.248	199.891	326.547	63.743
02	58.625	25.647	214.623	43.423	78.519	34.829
03	0.493	2.481	42.648	12.341	2.438	7.257
04	0.573	2.573	6.775	2.286	2.382	1.967
05	5.380	6.415	96.519	41.629	23.065	21.698
06	11.837	7.696	17.509	7.270	2.883	9.577
07	10.911	9.100	25.138	9.346	8.592	8.072
08	12.529	16.729	325.808	79.576	16.878	33.347
09	5.562	16.785	31.988	16.006	31.190	14.703
10	4.677	22.755	28.935	9.474	11.387	11.057

**Table 3 sensors-22-01940-t003:** Relative trajectory errors of different algorithms under different sequences.

Dataset	Ours	SuperPoint	ORB	SIFT	FAST
01	0.317	0.357	1.869	0.861	0.906
02	0.275	0.1455	0.808	0.157	0.260
03	0.102	0.073	0.180	0.111	0.090
04	0.061	0.056	0.172	0.085	0.057
05	0.057	0.059	0.469	0.135	0.120
06	0.125	0.084	0.163	0.072	0.058
07	0.083	0.093	0.168	0.092	0.082
08	0.109	0.122	0.796	0.243	0.148
09	0.120	0.139	0.183	0.153	0.153
10	0.145	0.168	0.136	0.092	0.145

## Data Availability

The data presented in this study are available upon request from the corresponding author.

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
