# Peer review of "DAN-SuperPoint: Self-Supervised Feature Point Detection Algorithm with Dual Attention Network"

_sensors, 2022, doi:10.3390/s22051940_

Round 1

Reviewer 1 Report

In this manuscript, the authors present a network designed to extract feature points in images in a self-supervised manner. The authors base their work on the Superpoint algorithm by DeTone et al. 

This reviewer has some minor concerns that should be corrected prior to publication:

1. Figure 1 is hard to understand due to a lack of labels. Please insert labels to properly describe the process (e.g., Pre-training, self-labeling etc.). Please also explain what each label (W, H, I, O P2, P1 F2 etc.) are in the caption of the figure.

2. Figure 2: The feature point results are incredibly difficult to assess. There are simply too many lines on the image and no proper explanation. Please reduce the number of points drawn on the image to allow readers to properly confirm the accuracy of the matching. What are the red lines in the SIFT results supposed to represent (presumably incorrect results)? Please explain in the caption.

3. The caption for figure 3 is missing the description for the right column of images. What are they supposed to be?

Reviewer 2 Report

The article is very interesting since the stated objective of improving feature point detection would help current and future developments. From my point of view the article should be improved in certain issues.

Introduction: there should be a description of the different sections of the article. 

Related Work: It is necessary to expand this section, not only bibliographical references, but also the explanations of the different authors. In my opinion it should not begin the sentences with the reference to the authors. 

Network Structure: What is a convolution kernel, it would be necessary to explain it and say what types there are and what is the reason to use the proposed one.

Section 4. It is necessary to explain or reference the ADAM optimizer justifying the choice of parameters.

Section 5. The conclusions are poor and a discussion section is needed.

Round 2

Reviewer 2 Report

The authors have followed the proposed recommendations and have significantly improved the work.